# Model Evaluation for Forecasting Traffic Accident Severity in Rainy Seasons Using Machine Learning Algorithms: Seoul City Study

**Jonghak Lee [1], Taekwan Yoon [2,\*] 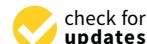, Sangil Kwon [1] and Jongtae Lee [1]**

[1] Transportation Pollution Research Center, National Institute of Environmental Research, (Environmental Research Complex), Hwangyeong-ro 42 Seo-gu, Incheon 22689, Korea; jonghack2000@korea.kr (J.L.); heatksi@korea.kr (S.K.); leelee@korea.kr (J.L.)

[2] Smart Infrastructure Center, Korea Research Institute for Human Settlements, 5 Gukchaegyeonguwon-ro, Sejong-si 30149, Korea

\* Correspondence: tyoon@krihs.re.kr; Tel.: +82-44-360-0394; Fax: +82-44-211-4775

**Abstract:** There have been numerous studies on traffic accidents and their severity, particularly in relation to weather conditions and road geometry. In these studies, traditional statistical methods have been employed, such as linear regression, logistic regression, and negative binomial regression modeling, which are the most common linear and non-linear regression analysis methods. In this research, machine learning architecture was applied to this problem using the random forest, artificial neural network, and decision tree techniques to ascertain the strengths and weaknesses of these methods. Three data sets were used: road geometry data, precipitation data, and traffic accident data over nine years corresponding to the Naebu Expressway, which is located in Seoul, Korea. For the model evaluation, three measures were employed: the out-of-bag estimate of error rate (OOB), mean square error (MSE), and root mean square error (RMSE). The low mean OOB, MSE, and RMSE observed in the results obtained using the proposed random forest model demonstrate its accuracy.

**Keywords:** machine learning architecture; random forest model; artificial neural network; decision tree algorithm; accident severity level; road surface condition; road hazard zone forecasting

## 1. Introduction

More than 3000 people die in road crashes every day worldwide, including approximately 1000 people in the United States. Such accidents cost $518 billion globally and $230.6 billion per year in the United States. In particular, many young people are affected by traffic accidents, as most traffic accidents are caused by the 15–29 age group (Association for Safe International Road Travel Data). Hence, substantial efforts have been made to address the main factors that cause traffic accidents, such as speeding, distracted driving, driver fatigue, road conditions, use of mobile phones, and poor weather conditions.

Regarding bad road conditions due to the weather, previous studies have revealed that there is a strong correlation between road friction coefficient and traffic accident risk [1,2]. The friction coefficient can be influenced by road conditions such as ice or wetness caused by snow or rainfall. Research has revealed that single-vehicle accidents are mostly caused by wet weather [3]. When a layer of water builds between the tire of a vehicle and the road surface, hydroplaning occurs. If all wheels experience hydroplaning simultaneously, the driver can no longer control the vehicle [4,5]. It has been shown that hydroplaning occurs at vehicle speeds of 80 km/h on thick water films when the water depth exceeds 2.5 mm [6]. According to the Korean Transportation Safety Authority, the total number of fatalities in

road traffic accidents in the country has decreased, but rain-related road deaths increased from 430 in 2013 to 463 in 2016. In particular, in Seoul, the traffic fatality rates on rainy days were the highest among all cities [7].

Traffic accidents are unpredictable and can happen anywhere and at any time. However, drivers can at least be provided with useful information to help avoid accidents or reduce their probability. The forecasting of traffic accidents and the identification of related factors under various conditions are important for preventing accidents and reducing their frequency. For these reasons, traffic accident prediction models have been developed to reveal the significant factors that affect traffic accidents so that traffic safety can be improved by controlling and/or improving these factors [8].

The rest of this paper is organized as follows. Next, we provide literature reviews on traffic accident forecasting and machine learning applications. The methodologies of the proposed random forest, artificial neural networks (ANN), and decision tree algorithms are introduced, and the data sets employed are described. The modeling results are presented as the estimated values (importance) of factors influencing accidents as well as interpretations of the model evaluations including the mean square error (MSE) and root mean square error (RMSE). Finally, a discussion is provided and conclusions are drawn.

## 2. Literature Reviews

### 2.1. Traffic Accident Forecasting Model

Earlier research has revealed that the difference between the speed of a vehicle and those of the other vehicles in the traffic flow [9] and reckless driving are factors that are strongly correlated with accidents [10,11]. Other studies have investigated the relationships between traffic accidents and the designed road geometry and weather conditions, especially rainfall, using statistical methods [12–16]. It is revealed that rain factors including water depth are strongly correlated with traffic accident severity using the structural equation model [17]. On the other hand, other studies revealed that the effect of precipitation does not affect directly to accident severity, but increasing accident frequency does [18,19].

Approaches including a zero-inflated hierarchical ordered probit and random parameter logit model were utilized to figure out the correlations. The models explain the relation better and show a reasonable level of accuracy from less detailed data [20,21].

In previous studies, the methods used for modeling traffic accidents were primarily statistical, including linear regression modeling, logistic regression modeling, and negative binomial regression modeling, which are the most common types of linear and nonlinear regression analysis. These methods not only involve strong assumptions but also are limited in their application as some are not always appropriate, such as when the outcome (the response variable) is discrete [22–24].

### 2.2. Machine Learning Applications

Machine learning is a well-developed technique and is being applied in many academic fields, although there are not many examples in transportation research. Previous studies have focused on the causes of traffic accidents and the severity of injuries using neural networks and decision trees [25], and recurrent neural network [26]. Several other studies have been performed to forecast fundamental traffic parameters including speed, volume, and density [27,28] through deep learning approaches. There was also an attempt to forecast bus ridership at the stop and stop-to-stop levels and vehicle traffic flow prediction including delay with the development of a deep learning architecture [29–31]. In addition, a rainfall prediction model was developed by a machine learning algorithm [32].

#### 2.2.1. Decision Tree

In recent studies, decision tree algorithms dealing with numerous non-parametric techniques in the field of data mining have been actively developed; these can solve complex relationships without

the need for any assumptions about the nature of the nonlinearity [33]. Decision tree algorithms are popular for various machine learning tasks. They are invariant under scaling and various other transformations of feature values and are robust against the inclusion of irrelevant features. However, their low accuracy remains a weakness [34].

In particular, trees that are grown deeply learn highly irregular patterns and tend to over-fit their training sets. In these cases, the bias is low but the variance is very high. Unlike tree models, the random forest model is a means of averaging multiple deep decision trees, trained on different parts of the same training set, with the objective of reducing the variance [34]. Furthermore, the use of the strong law of large numbers indicates that the trees always converge (if the mean is finite); therefore, there is no issue with over-fitting the data [35].

### 2.2.2. Random Forest

The random forest technique applies the bootstrap aggregation (bagging) approach to overcome the bias–variance trade-off, which was not completely solved in the previous approaches. Bias and variance are the factors that constitute learning error. When the bias is high, the forecasted results are not accurate compared to the actual results, and when the variance is high, the forecast fits very well for certain data sets but not for others, which indicates low forecasting stability. Bagging is a method of maintaining low bias and reducing high variance by sampling observations and features randomly in the training set. The random forest algorithm is the most popular one using the bagging approach; its accuracy (low bias) remains as an average value of a decision tree, and the variance is reduced by the use of the central limit theorem.

### 2.2.3. Artificial Neural Networks

Artificial neural networks (ANNs) have previously been used in many fields for prediction as computer models designed for knowledge processing [36,37]. ANNs can be used in various powerful ways: to teach non-linear system dynamics without any mathematical modeling [38–40], to analyze and generalize from samples and to make predictions, to memorize the features of given data, and to match or make associations from new data to old data [41]. In transportation research, many studies have applied ANN models to predict traffic accidents and severity [42–44], and such models have shown greater accuracy in predicting injury severity compared to other traditional methods, including ordered probit modeling [45]. On the other hand, ANN is disadvantageous for interpreting human factors to predict traffic accident severity [26].

In this study, traffic accident severity level prediction models were established with several parameters using three algorithms (random forest, ANN, and decision tree algorithms). Each model was successfully applied to predict traffic accidents based on the road surface conditions on a data set with numerous traffic accidents.

## 3. Methodology

This section reviews the random forest, ANN, and decision tree algorithms and discusses their application for forecasting traffic accident severity. The three algorithms were implemented using the statistical software ***R*** in this study.

### 3.1. Random Forest

The random forest model is a widely used extension of bagged trees [33]. After a random forest is trained on data, the tree functions make predictions. Figure 1 illustrates the structure of the random forest model. The classification "votes" are compiled and can be used to classify the given point based on the consensus "winner," or the vote distribution can be used to derive a probability for each possible class, as shown in Figure 1.

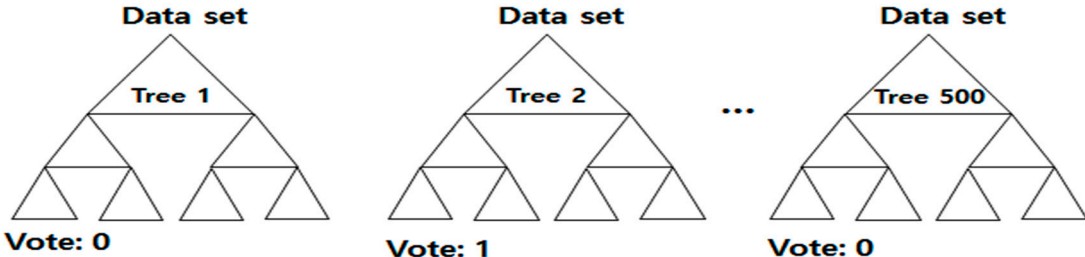

**Figure 1.** Conceptual diagram of random forest model.

The random forest training algorithm applies the bagging technique. Bagging, a shorthand term for "bootstrap aggregation," is an ensemble approach based on resampling or bootstrapping [46]. Instead of using the error from the prior round of fitting as a dependent variable or weight in subsequent rounds of fitting, bagging uses numerous random samplings of records in the data to fit many trees. In this study, the training examples were $Z_1 = (x_1, y_1), \ldots, Z_n = (x_n, y_n)$, an input $x$ was applied to a prediction problem, and a base learner $\hat{\theta}(x) = t\left(x; Z_1^*, \ldots, Z_n^*\right)$ was utilized. If $t(x; Z_i)$ is a decision tree trained on data, then the quantity $\theta(x)$ would be the output of the tree predictor on input $x$. Using bagging, it was attempted to stabilize the base learner $t$ by resampling the training data. The bagged version of $\hat{\theta}(x)$ is defined as follows:

$$\hat{\theta}^\infty(x) = E_*\left[t\left(x; Z_1^*, \ldots, Z_n^*\right)\right], \tag{1}$$

where the $Z_i^*$ series corresponds to sampling with replacement, which can be driven independently from the original data in the process of applying the bootstrap method. The expectation $(E_*)$ was obtained in terms of the bootstrap measure. Because its value could not be evaluated exactly, the Monte Carlo bagged estimator was used. In Equation (2), $Z_{bi}^*$ is an element $i$ in the $b$th bootstrap sample. Here, $B$ is defined as the number of trees grown.

$$\hat{\theta}^B(x) = \frac{1}{B}\sum_{b=1}^{B} t_b^*(x), \text{ where } t_b^*(x) = t\left(x; Z_{b1}^*, \ldots, Z_{bn}^*\right). \tag{2}$$

*3.2. ANN*

There are numerous types of ANNs, of which the most commonly used type is the multilayer perceptron (MLP), which consists of three layers (input, hidden, and output), each of which has nodes and activation functions [43]. In this study, a neural network with MLP architecture was designed and implemented using the **R** package program.

The schematic structure of the MLP in the ANN model is presented in Figure 2. An MLP in an artificial neuron that consists of inputs $(x_i)$, weights $(\omega_i)$, a summing function $(\Sigma)$, an activation function $(f)$, threshold value $(\theta)$, and an output $(y)$. The mathematical expression of the MLP is defined as follows:

$$y = f\left(\sum_{i=1}^{N} \omega_i x_i + \theta\right). \tag{3}$$

In the basic ANN cell shown in Figure 2, $x_1$, $x_2$, $\cdots x_n$ denote the cell inputs. The obtained inputs are multiplied times the weights $(\omega)$, and the net input is formed by summing with threshold values varying between $-1$ and $+1$.

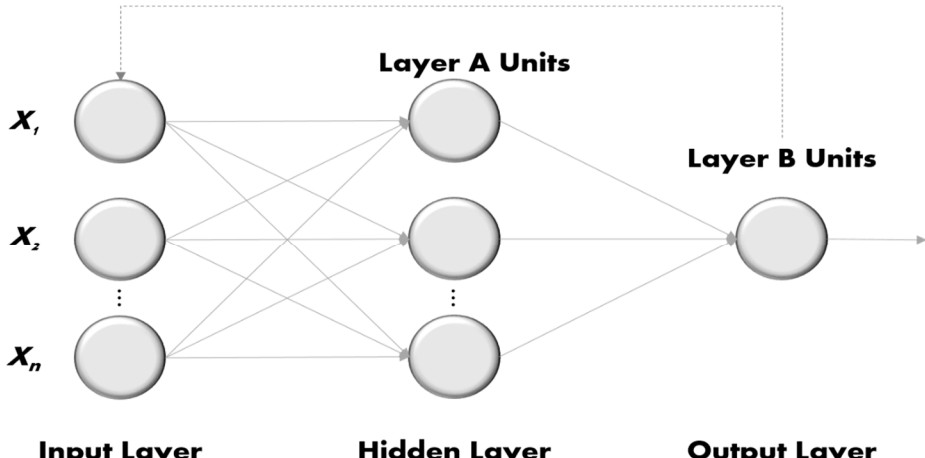

**Inputs**

**Activation function (Sigmoid)**

Σ

**Summing function**

**f**

y

**Weight**

**Figure 2.** Schematic diagram of multilayer perceptron (MLP).

The role of activation functions is to make a neural network non-linear. These functions are useful for binary classification schemes. The generally preferred activation functions include the tangent hyperbolic, sigmoid, and linear functions. It is desirable to have a continuous and differentiable activation function [47,48]. In this study, the sigmoid function was adopted, which is mostly used to generalize within an ANN and is defined as follows:

$$x = \frac{1}{1 + e^{-x}}. \tag{4}$$

Training was conducted using the backpropagation (BP) algorithm in the *R* package program. This method is utilized to compute the gradients of loss functions regarding the weights in ANNs. It is most widely used for the optimization of network performance by adjusting the weights. BP training consists of three steps [49,50]. The design utilized in this study is depicted in Figure 3.

**Layer A Units**

**Layer B Units**

$X_1$

$X_2$

$X_n$

**Input Layer**

**Hidden Layer**

**Output Layer**

**Figure 3.** Artificial neural networks (ANN) design in this study.

### 3.3. Decision Tree

The decision tree model known as a classification and regression tree (CART) [51] is a popular data mining approach that is applied in traffic safety studies [52,53]. The CART model is utilized for

comparison with statistical regression models and does not require any prespecified functional form, variable transformation, probability distribution, or error terms for fitting [54].

The CART model consists of a hierarchy of univariate binary decisions like an inverted tree, which grows from the top down. Each internal node in the tree specifies a binary test on a single variable, each branch represents an outcome of the test, and each leaf node represents a class label or class distribution.

A CART chooses the most suitable variable to split the data into two groups at the root node, partitioning the data into two disjoint branches such that the class labels in each branch are as homogeneous as possible; then, splitting is recursively applied to each branch, and so forth [55]. For instance, if dataset $T$ contains examples from $n$ classes, the Gini index ($gini(T)$) is defined as follows, where $pj$ is the relative frequency of class $j$ in $T$:

$$gini\ (T) = 1 - \Sigma j = 1 \text{ to } n\ pj^2. \tag{5}$$

If dataset $T$ is divided into two subsets $T1$ and $T2$ with sizes $N1$ and $N2$, the Gini index of the split data contains examples from $n$ classes, $gini(T)$ is defined as follows:

$$gini \text{ split } (T) = N1/N\ gini(T1) + N2/N\ gini(T2). \tag{6}$$

Then, CART exhaustively searches for univariate splits. The attribute providing the smallest $gini$ split $(T)$ is chosen to split the node. CART recursively expands the tree from a root node and then gradually prunes back the large tree.

## 4. Data Descriptions

### 4.1. Location and Road Geometry Data Set

To analyze traffic accidents on an expressway, various types of geometric conditions are required. After reviewing several highways in Seoul, the Naebu Expressway, one of six motorways, was selected as the most appropriate site for this study (Figure 4). Its length is 17.5 km (10.9 mi) (excluding the tunnel), and it is the motorway with the highest severity rate under wet conditions. The available information about the expressway includes geometric characteristics such as the vertical grade, super-elevation, curve radius, road type (tangent or curve), and curve length.

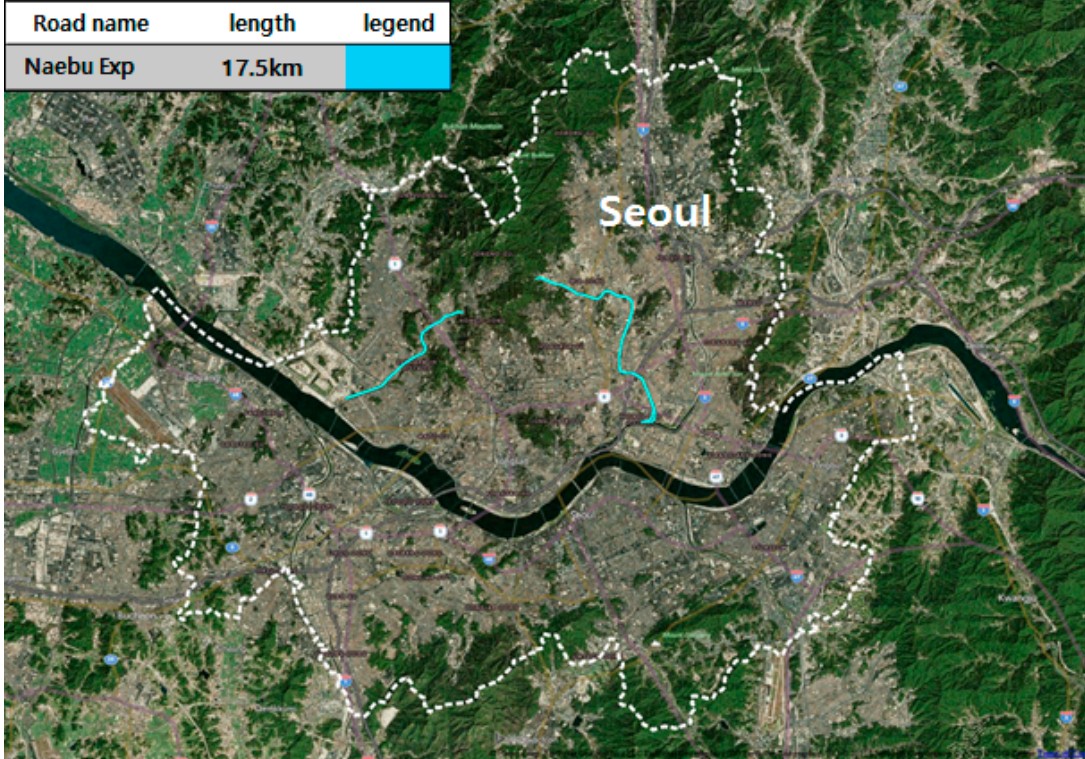

**Figure 4.** Research site map.

### 4.2. Traffic Accident Data Set

All reported accident data in Korea can be obtained from the Traffic Accident Analysis System. A sample of 518 subjects was acquired for accidents on the Naebu Expressway for the nine-year period from 2007 to 2015. The data contained information on accident location (longitude and latitude), date, time, driver characteristics (age and gender), and road surface conditions (dry and wet).

### 4.3. Rainfall Data and Rain Estimation

Rainfall information was obtained from the Korea Meteorological Administration. Then, the rainfall intensity on the road was estimated based on the rainfall data obtained from six weather stations selected from the 30 weather stations in Seoul (Figure 5). The K-nearest neighbor methodology, one of the most common classification algorithms used to predict the class of a record with an unspecified class based on the classes of its neighbor records [56], was adopted to estimate the rainfall intensity. The average, maximum, and minimum distances between the six weather stations and the research site are 1.9 km (1.2 mi), 3.38 km (2.10 mi), and 0.29 km (0.18 mi), respectively. Snow data were excluded from this study because of the difficulty of collecting them.

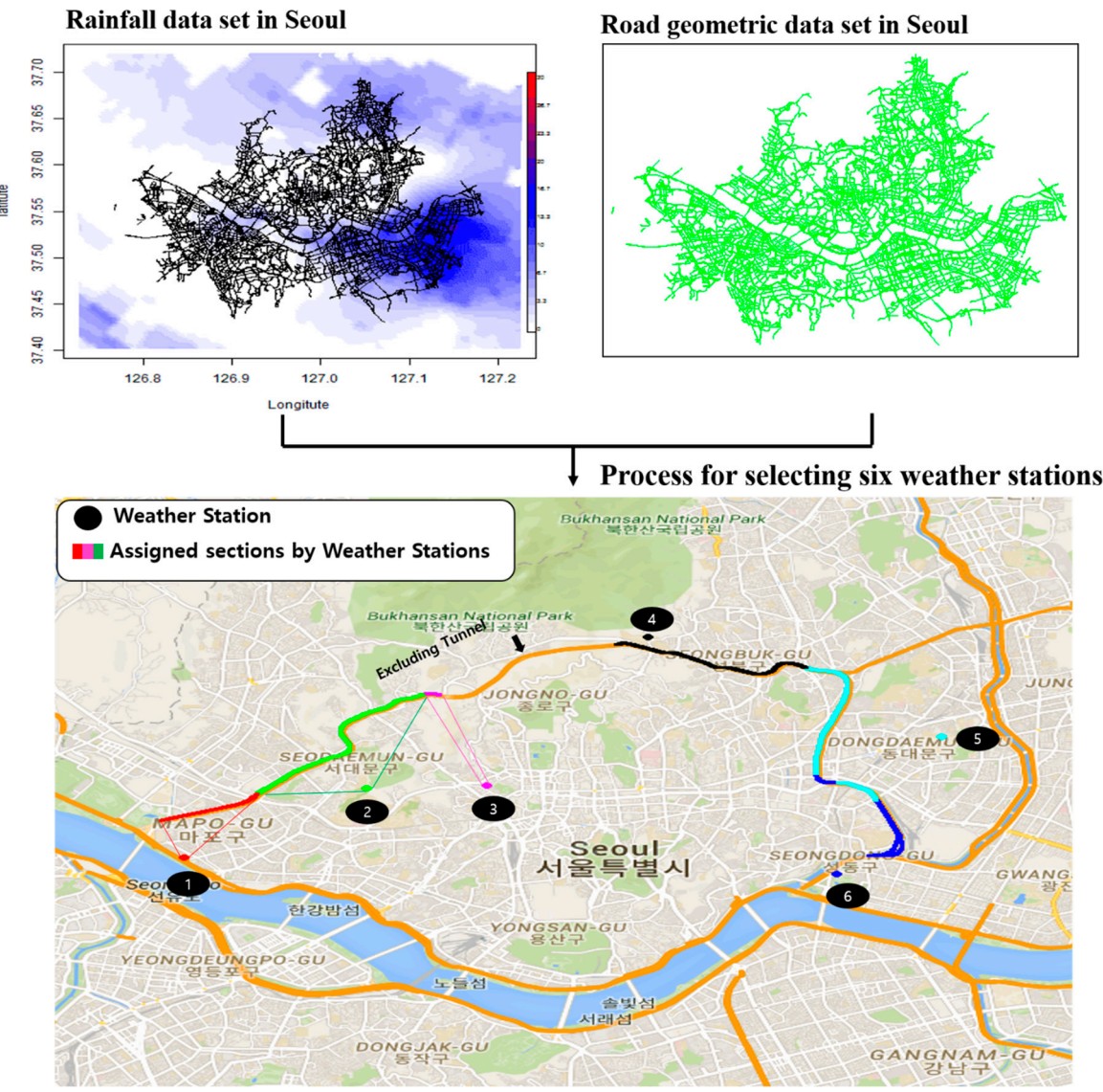

**Figure 5.** Process for estimating rainfall on the Naebu Expressway.

## 4.4. Variable Classification

Descriptions of the variables are provided in Table 1. The data used in this study were derived from a sample of 518 subjects involved in fatal, serious injury, and minor injury accidents reported for the Naebu Expressway in traffic police records. The variable accident level (AL) has two levels: non-critical incident (0) and critical incident (1). The criterion for traffic accident severity in Korea has weights of 1.2 for each fatality and 0.3 for each non-fatal injury [57]. A critical incident (1) in this study had a weight of at least 1.2. In other words, a case with a minimum of one fatality or four injuries was considered a critical incident. Thus, an accident was classified as a non-critical incident (0) if it resulted in at least one injury but no fatalities and as a critical incident (1) if at least one fatality or four or more injuries resulted from the accident. Among the explanatory variables (independent variables), the road alignment factors (road geometry type (RT), vertical grade (VS), super-elevation (Se), horizontal alignment (HA), and curve length (CL)), rainfall intensity (RI), and driver age (Da) were continuous variables, and the others were categorical variables. The road surface condition (RC) was divided into two levels (dry and wet).

**Table 1.** Variable descriptions.

| Classification | Abbreviation | Description | Category Coding/Value |
|---|---|---|---|
| Accident | AL | Accident level | Range of 0.08–1.88 <br> *One fatality has a weight of 1.2* <br> *one non-fatal injury has a weight of 0.3* |
| Road alignment factors | RT | Road geometry type | 0 = Curve <br> 1 = Tangent |
| | VS | Vertical grade | Range of –4.3–10.8% <br> *Mean*: 0.01% |
| | Se | Super-elevation | Range of –4.1–4.1% <br> *Mean*: 0.01% |
| | HA | Horizontal alignment | Range of 180–3550 m <br> *Mean*: 625 m |
| | CL | Curve length | Range of 180–760 m <br> *Mean*: 393 m |
| Weather factors | RI | Rainfall intensity | Range of 0.5–56 mm/h <br> *Mean*: 3.7 mm/h |
| | RC | Road surface condition | 0 = Dry <br> 1 = Wet |
| Road environment factors | VT | Vehicle type | 1 = Small passenger car <br> 2 = Large passenger car <br> 3 = Other |
| | AT | Accident time | 0 = Night <br> 1 = Day |
| Human factors | Da | Driver age | Range of 19–81 years <br> *Mean*: 43 years |
| | Dg | Driver gender | 0 = Male <br> 1 = Female |

NOTE: 1 m = 3.28 ft; 1 mm = 0.0394 in.

## 5. Modeling Results

### 5.1. Data Set

In the models, the original data set (a sample of 518 subjects, as described in the previous section) was divided into two parts: a training data set and a test data set. The model was developed by training the classifier using the training data set, and the performance of the classifier was tested on the test data set. For this study, 75% of the data were utilized in the training data set, and the rest were used as the test data set. Since the process of assigning data can influence the model outcomes, the cases were randomly assigned to the training and test data subsets.

Before evaluating the models, the out-of-bag estimate of error rate (OOB) of each model was calculated to identify the optimal number of trees for model evaluation. The OOB is the mean prediction error on each training sample $x_i$ determined using only the trees that do not have $x_i$ in their bootstrap sample [58].

In this study, 500 trees were used and were divided into samples of five sizes, $N$ = {100, 200, 300, 400, 500}, for simulation. For each data point $(x_i, y_i)$, the classifiers built on a bootstrap sample that did not contain $(x_i, y_i)$ were retained and aggregated. Then, the predicted labels were compared to obtain the final value, $y_i$. After performing these steps for each data point $(x_i, y_i)$ of the learning set, the OOB for each point was calculated as the mean of the OOBs over 10 runs to avoid insignificant-sampling effects. The resulting OOB was 4%. Figure 6 illustrates the OOB, showing that both curves tend to stabilize when the number of trees exceeds 200.

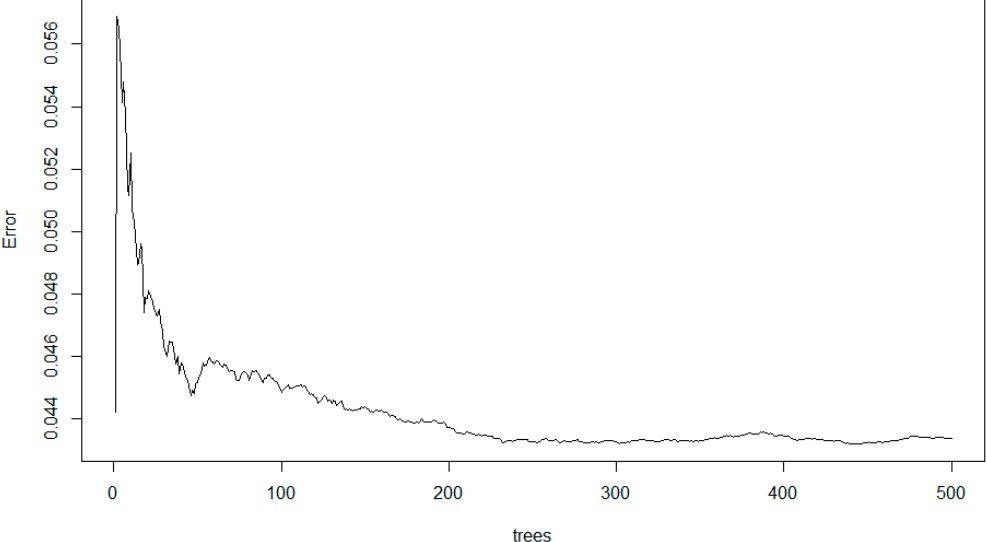

**Figure 6.** Mean out-of-bag estimate (OOB) over 10.

*5.2. Model Evaluation*

To evaluate the model performance, the difference between the predicted and true values of every incident was calculated. The prediction values for AL were obtained from the individual models using the test data set (sample of 130 subjects). The random forest model yielded the closest values to the observed values among the three models, while the ANN model tended to overestimate AL compared with the observed values. Previous research has also indicated this overestimation issue in ANN prediction models [59]. The results are presented in Table 2.

**Table 2.** Predicted accident level (AL) values obtained from three models.

| Number | Observed Values | Predicted Values | | |
|:---:|:---:|:---:|:---:|:---:|
| | | **Random Forest** | **ANN** | **Decision Tree** |
| 1 | 0.08 | 0.19 | 0.47 | 0.49 |
| 2 | 0.28 | 0.16 | 0.43 | 0.49 |
| : | : | : | : | : |
| 50 | 0.2 | 0.20 | 0.47 | 0.47 |
| 51 | 0.4 | 0.23 | 0.47 | 0.47 |
| 52 | 0.08 | 0.28 | 0.80 | 0.55 |
| 53 | 0.2 | 0.21 | 0.43 | 0.47 |
| 54 | 0.16 | 0.22 | 0.43 | 0.24 |
| 55 | 0.40 | 0.25 | 0.80 | 0.38 |
| : | : | : | : | : |
| 129 | 0.2 | 0.14 | 0.20 | 0.55 |
| 130 | 0.36 | 0.19 | 0.47 | 0.43 |
| Mean | 0.21 | 0.22 | 0.46 | 0.37 |

In the examples in Figure 7, the blue lines show the predicted outputs of the AL values obtained from the three models on Seoul roads. Overall, the outputs of each model do not differ significantly, although the AL values correspond to different levels. In particular, the ANN has more blue lines on the roads, while the random forest model tends to have the fewest blue lines on the roads, as shown in Figure 7.

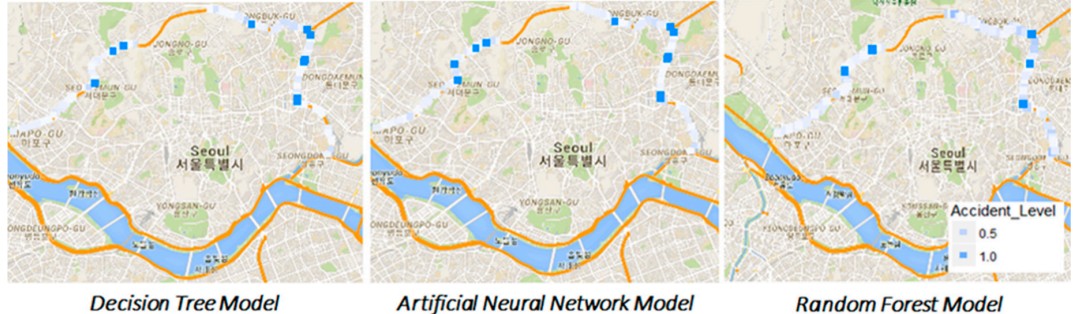

**Figure 7.** Examples of predicted AL values obtained from the three models.

The performance of the models was evaluated in detail using the MSE and RMSE, which were calculated for the test data. Low values of MSE and RMSE indicate that a model can be considered superior. These quantities are defined as follows:

$$\text{MSE} = \frac{1}{N} \sum_{i=1}^{n} (t_{mi} - t_{gi})^2, \quad \text{RMSE} = \sqrt{\frac{1}{N} \sum_{i=1}^{n} (t_{mi} - t_{gi})^2}. \tag{7}$$

For model evaluation, $t_{mi}$ is the $i$th observation value, $t_{gi}$ is the $i$th model value, $N$ is the number of trained data, and $m$ is the number of parameters in the model (the total number of weights and invariables in the net structure). The MSE and RMSE values for the models are shown in Table 3. The lower MSE and RMSE of the random forest model indicate its superior performance among the three models. Meanwhile, the ANN model exhibits higher values of MSE and RMSE, which is assumed to be because the ANN model overpredicted the AL.

**Table 3.** Model comparison results.

| Model | MSE | RMSE |
| --- | --- | --- |
| Random Forest | 0.047 | 0.217 |
| ANN | 0.102 | 0.319 |
| Decision Tree | 0.079 | 0.281 |

## 6. Case Study for Forecasting Traffic Accident Severity

*Variable Importance*

The variable importance plot is a crucial output of the random forest model. For each variable in the matrix, this plot presents the importance of the variable in classifying the data. The plot shows each variable on the *y*-axis and its importance on the *x*-axis. These variables are ordered top to bottom from the most to least important. The mean decrease in accuracy is how much the model fit decreases when a variable is dropped. The greater the drop, the more significant the variable.

The top two feature variables in the random forest importance list (Table 4) are the RI, which is a meteorological factor, and curve length (CL), which is a road alignment factor. These results presume the probability of traffic accidents to be higher when driving into a curve in the rain. The feature ranked third is the gender of the driver (Dg), which is a human factor. These three top-ranked variables for AL are evenly distributed among the three factors (weather, road alignment, and human). Table 4 shows the importance values for the feature variables in both models.

**Table 4.** Importance values (mean decrease in accuracy).

| Feature Variable | Mean Decrease in Accuracy | |
| --- | --- | --- |
| RI (rainfall intensity) | 1.6878244076 | 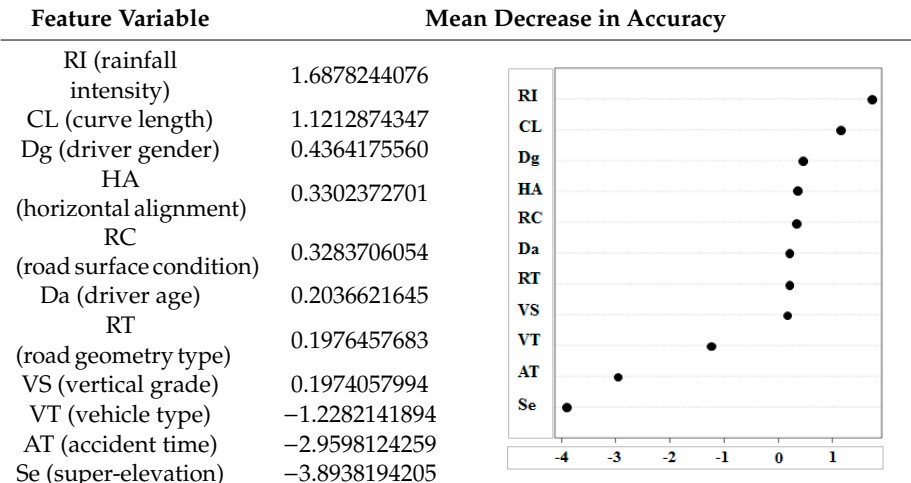 |
| CL (curve length) | 1.1212874347 | |
| Dg (driver gender) | 0.4364175560 | |
| HA (horizontal alignment) | 0.3302372701 | |
| RC (road surface condition) | 0.3283706054 | |
| Da (driver age) | 0.2036621645 | |
| RT (road geometry type) | 0.1976457683 | |
| VS (vertical grade) | 0.1974057994 | |
| VT (vehicle type) | −1.2282141894 | |
| AT (accident time) | −2.9598124259 | |
| Se (super-elevation) | −3.8938194205 | |

## 7. Conclusions

In this study, random forest, ANN, and decision tree algorithms were applied to road accident data. The random forest model yielded the most accurate predictions, while the ANN algorithm overestimated the AL compared to the other models. In the model evaluation using the MSE and RMSE, the ANN model produced the highest values due to AL overestimation.

After the random forest model was verified to be superior for prediction, variables having strong correlations with traffic accidents were analyzed and an accident forecast model was built using the random forest algorithm. The model was developed to estimate accident severity according to road surface conditions using traffic accident data.

Three data sets containing road geometry data, precipitation data, and traffic accident data for the Naebu Expressway in Seoul, Korea over the course of nine years were used. The data were analyzed using the random forest algorithm as a machine learning tool because this algorithm can handle complex relationships without any assumptions and reduces the risk of over-fitting, which is the main disadvantage of tree algorithms; furthermore, it is possible to evaluate various traffic accident characteristics. In the model evaluation, the low mean error of the OOB and low values of the MSE and RMSE demonstrate the accuracy of the proposed model.

From the variable importance plot, the RI, CL, and Dg are the top featured factors, which can be considered to be important parameters affecting the number of accidents based on the road surface conditions for the Naebu Expressway. This finding supports previous literature reports, where the main factor affecting traffic injury severity was determined to be the designed road geometry [60,61] and one of the factors that was significantly related to severity was road wetness [62,63]. In addition, it has been found that the traffic accident severity correlates with Dg and that male drivers show increased severity [64].

This work will improve the ability of decision-makers such as policy-makers and transportation safety designers to take proper actions for traffic safety control of Seoul under various weather conditions. Furthermore, this study will enable researchers adopting the random forest model for traffic accident forecasts to obtain more accurate results.

Even though the accident data covered a nine-year period from 2007 to 2015 for the Naebu Expressway, the low quantity of data for particular road surface conditions did not allow an analysis of wet road conditions. In further research, larger data sets will allow estimation for the various types of road conditions, which is expected to improve the prediction performance. A limitation of the current study is that the analysis does not consider about the road construction situation, which causes the same amount of rain to have different consequences, but the assumption that the same amount of rain has the same consequences.

**Author Contributions:** Conceptualization, J.L. (Jonghak Lee). and T.Y.; methodology, T.Y.; software, J.L. (Jonghak Lee); validation, S.K.; formal analysis, J.L. (Jongtae Lee); writing—original draft preparation, J.L. (Jongtae Lee); writing—review and editing, T.Y.; visualization, S.K.; supervision, J.L. (Jonghak Lee), All authors have read and agreed to the published version of the manuscript.

**Funding:** This research was funded by National Institute of Environmental Research (NIER), Ministry of Environment (MOE), grant number NIER-2018-01-01-072.

**Conflicts of Interest:** The authors declare no conflict of interest.

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
