# Peer review of "Model Evaluation for Forecasting Traffic Accident Severity in Rainy Seasons Using Machine Learning Algorithms: Seoul City Study"

_applsci, doi:10.3390/app10010129_

Round 1

Reviewer 1 Report

In row 37 there should be "friction coefficient" instead of "friction" The quality of figures should be improved. In their current form they are illegible (e.g. Figure 5, Figure 7) In Figure 2, the weights of individual criteria are marked with the letter "w" while in the text is a small omega letter (row 146) In line 146, the letter "i" should be in the subscript In the formula (3) there is the Greek letter theta, which was not explained in the text of the paper Formula numbers (5) and (6) are not in the same row as the formulas The paper should contain explanations regarding the adopted research methodology in the scope of rainwater preservation on a specific section of the road. Due to the road construction, the same amount of rain can have different consequences for the road surface and thus its friction coefficient Incorrect letter case in chapter 5 title Conclusions in their current form cannot be accepted and need to be changed. They are too obvious and not revealing enough. The main conclusion drawn from the research is the impact of rainfall intensity, curve length and driver gender on the number of accidents. It seems that conducting complex research to achieve such obvious conclusions is not quite the right approach            

Author Response

We graciously thank you for your substantive and constructive comments. These comments have motivated a great deal of work, which we believe has significantly improved the paper. Below, we have listed each of the comments along with our replies. 

In row 37 there should be "friction coefficient" instead of "friction" 

-> We have revised it.

The quality of figures should be improved. In their current form they are illegible (e.g. Figure 5, Figure 7)

-> We have revised Figure 5 and improved Figure 7 resolution.

In Figure 2, the weights of individual criteria are marked with the letter "w" while in the text is a small omega letter (row 146)

-> We have revised it.

In line 146, the letter "i" should be in the subscript

-> We have revised it.

In the formula (3) there is the Greek letter theta, which was not explained in the text of the paper

-> We have added the explanation of theta, threshold value. 

Formula numbers (5) and (6) are not in the same row as the formulas

-> We have revised it.

The paper should contain explanations regarding the adopted research methodology in the scope of rainwater preservation on a specific section of the road. Due to the road construction, the same amount of rain can have different consequences for the road surface and thus its friction coefficient

-> Yes, we agree on this. The authors added this in limitation section. 

Incorrect letter case in chapter 5 title Conclusions in their current form cannot be accepted and need to be changed.

-> We have revised it.

They are too obvious and not revealing enough. The main conclusion drawn from the research is the impact of rainfall intensity, curve length and driver gender on the number of accidents. It seems that conducting complex research to achieve such obvious conclusions is not quite the right approach   

-> Road is the concept of 'two dimension, line'. Along with this, we apply weather data and integrate with machine learning method. This causes complex research. We agree on this comment and believe that using this new methodology contributes academic achievement. 

Reviewer 2 Report

The paper reads well and provides results from RFs, ANNs and Decision Trees for road accident severity prediction. The overall methodology is well explained and the explanatory analysis of the most important predictors are presented in a detailed way. A crucial aspect that is missing, is the positioning of the work with regards to the state of the art in the literature. Are the results comparable to other work in the literature? Are the results better/worse than previous work? What could be enhanced? Why weren't other methods discussed in the literature tested? 

Author Response

We graciously thank you for your substantive and constructive comments. These comments have motivated a great deal of work, which we believe has significantly improved the paper. Below, we have listed each of the comments along with our replies. 

The paper reads well and provides results from RFs, ANNs and Decision Trees for road accident severity prediction. The overall methodology is well explained and the explanatory analysis of the most important predictors are presented in a detailed way. A crucial aspect that is missing, is the positioning of the work with regards to the state of the art in the literature. Are the results comparable to other work in the literature? Are the results better/worse than previous work? What could be enhanced? Why weren't other methods discussed in the literature tested? 

-> As we mentioned in the main text, previous research works have studied the relationship between weather data and traffic accident using traditional methods such as regression model. However, there are limitations to deal real-time big data.

The analyzed data is real-time based so that we could not apply regression method to compare the result and reveal that the proposed method is better. Instead of this, we present the predicted values among three machine learning models, Random Forest, ANN, and Decision Tree.

Therefore, the proposed machine algorithm aims to process big data and improve the accuracy. This will contribute to predict traffic accident risk with high accuracy and reduce the accidents on rainy days. 
